# The cost of obtaining rewards enhances the reward prediction error signal of midbrain dopamine neurons

Shingo Tanaka[1], John P. O'Doherty[2,3] & Masamichi Sakagami[1]

Midbrain dopamine neurons are known to encode reward prediction errors (RPE) used to update value predictions. Here, we examine whether RPE signals coded by midbrain dopamine neurons are modulated by the cost paid to obtain rewards, by recording from dopamine neurons in awake behaving monkeys during performance of an effortful saccade task. Dopamine neuron responses to cues predicting reward and to the delivery of rewards were increased after the performance of a costly action compared to a less costly action, suggesting that RPEs are enhanced following the performance of a costly action. At the behavioral level, stimulus-reward associations are learned faster after performing a costly action compared to a less costly action. Thus, information about action cost is processed in the dopamine reward system in a manner that amplifies the following dopamine RPE signal, which in turn promotes more rapid learning under situations of high cost.

[1] Brain Science Institute, Tamagawa University, 6-1-1 Tamagawagakuen, Machida, Tokyo 194-8610, Japan. [2] Division of the Humanities and Social Sciences, California Institute of Technology, 1200 E California Blvd, Pasadena, CA 91125, USA. [3] Computation and Neural Systems, California Institute of Technology, 1200 E California Blvd, Pasadena, CA 91125, USA. Correspondence and requests for materials should be addressed to M.S. (email: sakagami@lab.tamagawa.ac.jp)

Humans and animals prefer a reward received after exerting a lot effort to obtain it compared to the same reward after a smaller amount of effort[1–3]. A number of explanations have been posited for this effect such as effort justification[4,5] and the contrast effect[6], in which greater value is attributed to an outcome obtained after paid effort. However, it remains unclear whether and how the processing of reward information in the brain is modulated by the effort expended to obtain a reward.

We focused specifically on the midbrain dopamine system, given the role of this system in promoting behavioral adaptation to rewards[7–9]. Dopamine neurons are known to represent reward prediction error (RPE) signals that can facilitate learning of reward predictions by the basal ganglia[10–17]. The strength of the RPE depends on the quantity, quality, and subjective value or utility of the reward[7,18–21]. Moreover, dopaminergic activity is modulated by costs and/or effort[22,23]. On this basis, we postulated that the dopaminergic RPE signal would be directly modulated by the cost paid to obtain a reward. Furthermore, because the RPE signal is causally involved in mediating learning of stimulus-reward associations[24–26], we hypothesized that the cost paid to obtain the reward would directly increase the learning speed of stimulus-reward associations.

To test our hypotheses, we measured both behavior and dopaminergic activity in two Japanese monkeys while they performed a saccade based effort task. Monkeys react faster to a reward-predicting cue that is presented after a high-cost (HC) action compared with that after a low-cost (LC) action. The activity of dopaminergic neurons to the reward-predicting cues are increased by the paid cost. In addition, learning speed to the stimulus-reward association is also enhanced by the paid cost. Therefore, we suggest that the cost paid to obtain rewards increases the RPE signal in dopamine neurons and thereby enhances stimulus-reward associations.

## Results

**High-Low cost (HLC) saccade task**. To examine the effect of paid cost on behavior and on dopamine neuron activity, the monkeys performed a saccade task with two cost conditions (Fig. 1a, see Methods). In HC trials, the monkeys quickly made a saccade to the target and held their gaze on it without blinking for a longer period (Fig. 1b; green lines). In contrast, on LC trials the monkeys looked around freely at first before fixating for a shorter period (Fig. 1b, purple lines). Because maintaining a long fixation is difficult for monkeys, they made more errors during delays on HC trials (Fig. 1c). To control for a consequent difference in reward probability between HC trials and LC trials, we inserted forced aborts in a portion of LC trials to equalize success rates and reward probabilities between trial types (Fig. 1d).

**Paid cost increases the value of reward-predicting cues**. To obtain implicit evidence for a difference in the monkey's subjective valuation of cues, we tested the monkeys' reaction times (RTs). In particular, we anticipated that if the monkeys assign a higher subjective value to one option than another, they should show faster RTs for the more valued option[27]. When RTs were compared between the cost cues, both monkeys showed faster RTs to the LC compared to the HC cue (Fig. 1e), demonstrating an implicit preference for the LC condition. When RTs were compared between reward cues, both monkeys showed faster RTs to reward (R+) cues than no reward (R−) cues (Fig. 1f), indicating that they preferred R+ cues to R− cues. In addition, both monkeys showed faster RTs to the $R_{HC}+$ cue compared with the $R_{LC}+$ cue and to the $R_{HC}-$ cue compared with the $R_{LC}-$ cue (Fig. 1f), indicating that they valued more the reward-predicting cues in the HC compared to the LC condition.

In addition, we included choice trials in the HLC saccade task to test monkeys' overt preferences between cues (Supplementary Fig. 1a). The monkeys preferentially chose the LC cue when they choosing between cost cues (Supplementary Fig. 1b). Monkey S preferentially chose the $R_{HC}+$ cue when performing the choice task between $R_{HC}+$ and $R_{LC}+$ cue, but no preference between $R_{HC}-$ and $R_{LC}-$ cues (Supplementary Fig. 1c, d). In contrast, monkey P showed no overt preference between $R_{HC}+$ and $R_{LC}+$ cues, but nevertheless preferred the $R_{HC}-$ cue when choosing between $R_{HC}-$ and $R_{LC}-$ cues (Supplementary Fig. 1c, d).

**Electrophysiological results in the HLC saccade task**. We recorded single unit activity from neurons located within the substantia nigra pars compacta (SNc) and ventral tegmental area (VTA) during the HLC saccade task. We identified 70 dopamine neurons across the two monkeys (Supplementary Fig. 2a; 18 and 52 neurons from Monkey P and S, respectively). Histological examination confirmed the neurons were located in or around the SNc/VTA (Supplementary Fig. 2b).

In Fig. 2, we show the activity of a representative dopamine neuron. This neuron showed modest activation to the LC cue, and phasic activation or suppression to the reward ($R_{HC}+$ and $R_{LC}+$) or no reward cues ($R_{HC}-$ and $R_{LC}-$), respectively (Fig. 2, Supplementary Fig. 3). The neuron also showed phasic activation to the unpredictable reward as well as phasic suppression in response to an aversive stimulus, an unpredictable air-puff (Fig. 2, right panel). In addition, the neuron showed modest suppression to the start cue but did no response to reward delivery. The whole population of dopamine neurons we recorded all showed similar responses to the start cue and reward delivery (Supplementary Fig. 4a, b). In the HLC saccade task, an effort cost was paid before obtaining the reward. Because the predicted cost reduced dopamine neuron activity[22,23], dopamine neuron responses would be suppressed at the time of start cue presentation[16].

**Dopamine neurons code information on both reward and cost**. The neurons responded phasically to the LC cue, but less robustly to the HC cue (Fig. 3a, b). Evoked responses to the cost condition cues exhibited a smaller response to the HC cue than LC cue (two-tailed Wilcoxon's signed-rank test, $P < 3.2 \times 10^{-4}$, $n = 70$). We quantified the effect of the predicted cost on neuronal responses using a receiver operating characteristic (ROC) analysis. The distribution of the area under the ROC curve (auROC) was significantly <0.5 (Fig. 3c; two-tailed Wilcoxon's signed-rank test; $P = 5.4 \times 10^{-4}$, $n = 70$), indicating that HC cue responses were smaller than to the LC cue. Predicted cost has previously been found to reduce dopamine neuron activity, consistent with our results[22,23]. In addition, the population of dopamine neurons exhibited significant activation to the LC cue while showing no significant suppression to HC cues (Supplementary Fig. 4c, d). These results suggest that dopamine neurons code and integrate information about both reward and cost at the time of cost cue presentation.

Two distinct subtypes of dopamine neurons have previously been described: motivational value and salience neurons[28,29]. We found evidence in our dopamine neuron population of response patterns consistent with both subtypes. Value type neurons showed phasic suppression to the aversive air-puff stimuli (Fig. 3d, e). Conversely, salience neurons showed a phasic activation to the aversive stimuli (Fig. 3g, h). The long fixation in the HC trial is also unpleasant and aversive; therefore, it is possible that the two dopamine neuron subtypes would show different response patterns to the cost condition cues. If dopamine neurons represent aversive stimuli and cost in a similar manner, then value neurons should show decreased

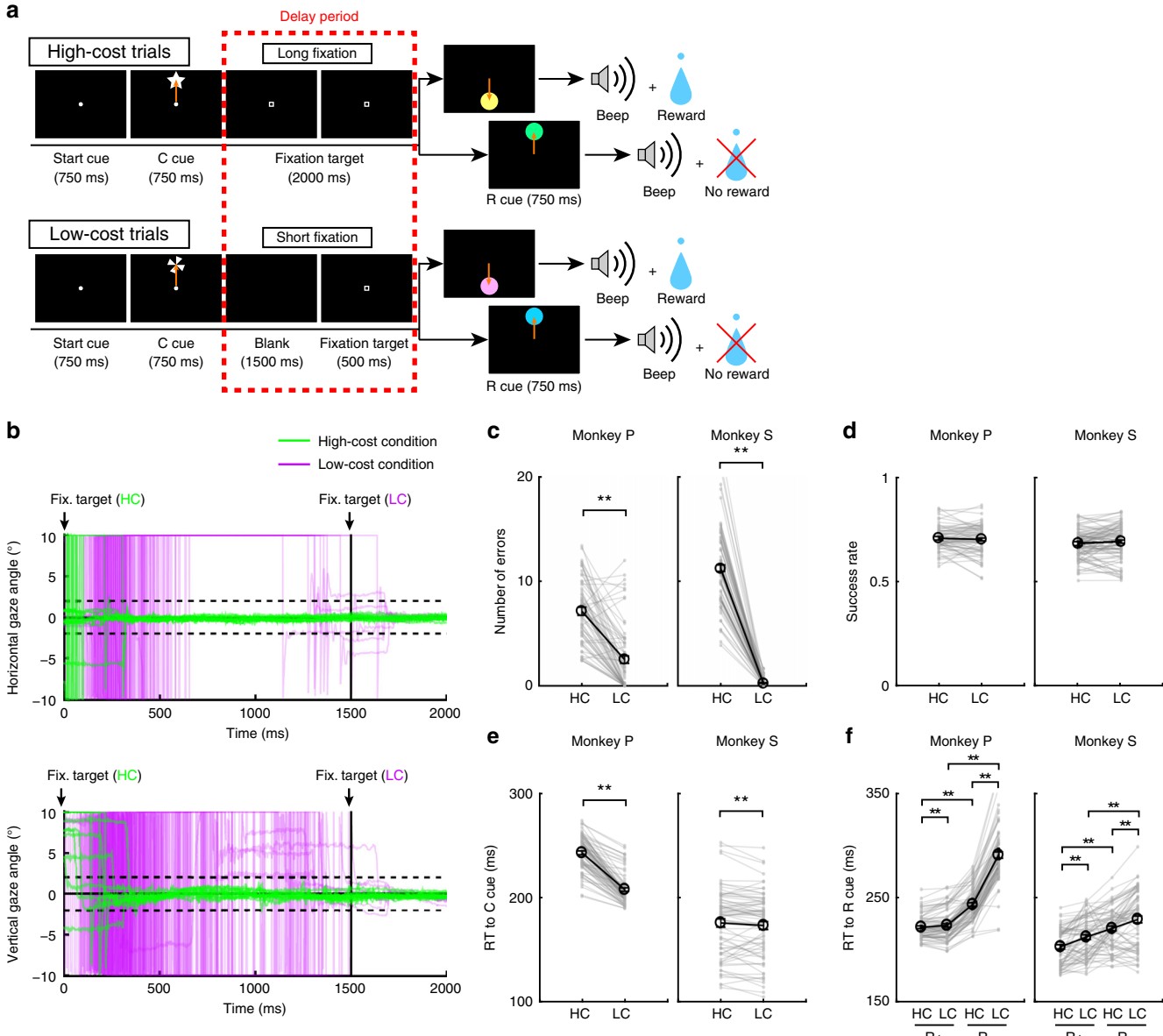

**Fig. 1** HLC saccade task. **a** The HLC saccade task. Cost cues (C cue) signaled the amount of effort that was required to achieve a potential reward. A long fixation was required during the delay period in high-cost trials. The reward cue (R cue) indicates whether the monkeys could obtain a reward or not. **b** The time course of the gaze angle during the delay period. The upper and lower panels show the horizontal and vertical gaze angles, respectively. Green and purple lines indicate the gaze angle in high-cost trials (50 trials in each panel) and in low-cost trials (50 trials in each panel), respectively. **c** The number of errors during the delay period in the high-cost and low-cost trials (**$P < 0.01$; two-tailed paired $t$ test; $t_{67} = 8.8$, $P = 4.8 \times 10^{-15}$, $n = 68$ for Monkey P; $t_{83} = 26.6$, $P \approx 0$, $n = 84$ for Monkey S). Black circles and error bars indicate mean and SEM. **d** Success rates in the high-cost and the low-cost trials (two-tailed paired $t$ test; $t_{67} = 0.51$, $P = 0.61$, $n = 68$ for Monkey P; $t_{83} = 0.79$, $P = 0.43$, $n = 84$ for Monkey S). **e** RTs to the cost cues (**$P < 0.01$; two-tailed paired $t$ test; $t_{67} = 20.4$, $P \approx 0$, $n = 68$ for Monkey P; $t_{69} = 2.0$, $P = 1.2 \times 10^{-3}$, $n = 70$ for Monkey S). **f** RTs to the reward cues (**$P < 0.01$; two-tailed paired $t$ test; Monkey P ($n = 68$): HC+ vs. LC+, $t_{67} = 3.5$, $P = 9.2 \times 10^{-4}$; HC− vs. LC−, $t_{67} = 24.5$, $P \approx 0$; HC+ vs. HC−, $t_{67} = 21.6$, $P \approx 0$; LC+ vs. LC−, $t_{67} = 28.5$, $P \approx 0$; Monkey S ($n = 70$): HC+ vs. LC+, $t_{69} = 5.6$, $P = 4.4 \times 10^{-7}$; HC− vs. LC−, $t_{69} = 4.8$, $P = 8.4 \times 10^{-5}$; HC+ vs. HC−, $t_{69} = 18.0$, $P \approx 0$; LC+ vs. LC−, $t_{69} = 5.9$, $P = 1.1 \times 10^{-7}$)

activity to the HC cue due to its aversiveness. On the other hand, salience neurons should increase in activity to the HC cue because they also increase to unpleasant stimuli. However, evoked responses of both types of neurons were smaller to the HC cue compared with the LC cue (two-tailed Wilcoxon's signed-rank test; $P = 0.021$, $n = 41$ and $P = 0.0044$, $n = 29$ for the value and the salience types, respectively), and the ROC analysis showed smaller responses to the HC compared to the LC cue in both subtypes (Fig. 3f, i; two-tailed Wilcoxon's signed-rank test; $P = 0.030$, $n = 41$ and $P = 0.0058$, $n = 29$ for the value and the

salience types, respectively). Thus, predicted cost reduced activity in both subtypes of dopamine neurons. These results indicate that cost information is processed by dopamine neurons in a qualitatively different way to aversive stimuli.

In the HLC saccade task, we inserted a forced abort in a portion of LC trials to equalize the success rates and the reward probability between trial types. This manipulation increased the uncertainty of obtaining reward or the risk of no reward in the LC condition. Therefore, the higher activity of dopamine neurons and the monkeys' increased valuation for the LC over the HC cue

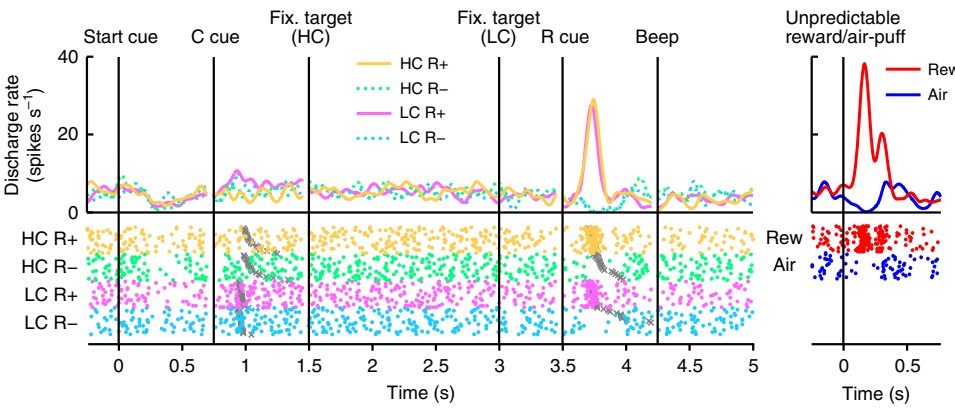

**Fig. 2** Activity of a representative dopamine neuron in the HLC saccade task. Spike density functions (convolved with a Gaussian function) and raster plots are aligned with the onset timing of the start cue, the cost cue (C cue), the fixation target, the reward cue (R cue), and the beep. Each color represents a condition (yellow: HC+, green: HC−, pink: LC+, cyan: LC−), respectively. The timings of the saccade onset are indicated by gray crosses. The responses of the dopamine neurons to the unpredictable reward or air-puff are also represented in the right panel (red: unpredictable reward, blue: unpredictable air-puff)

could be due to the difference in the risk or uncertainty between the cost conditions. However, we found no relationship between the number of forced aborts and the difference in RTs (Supplementary Fig. 5a, b), and we found a positive correlation between the number of forced aborts and the auROC (Supplementary Fig. 5c). We also compared dopamine responses to the cost cues after aborts vs. after correct trials, but found no difference in both cost conditions (Supplementary Fig. 5d). These results indicate that the number of forced aborts in the LC condition do not explain either the increase in valuation or the enhanced dopaminergic activation to the LC cue.

**Increased dopamine responses to reward cue by paid cost**. The recorded dopamine neurons were found to show phasic activation and suppression to reward and no reward-predicting cues, respectively (Fig. 2). Next, we assessed whether these responses were modulated by the cost previously incurred. An example of a representative neuron and population-averaged neurons exhibiting larger activation to the $R_{HC}+$ cue than the $R_{LC}+$ cue are shown in Figure 4a and b, respectively. (two-tailed Wilcoxon's signed-rank test; $P = 7.4 \times 10^{-5}$, $n = 70$). The distribution of auROCs was >0.5, indicating that the response to the $R_{HC}+$ cue was larger than to the $R_{LC}+$ cue (Fig. 4c; two-tailed Wilcoxon's signed-rank test; $P = 1.4 \times 10^{-4}$, $n = 70$). These results indicate that the response to the reward-predicting cue in the HC condition is significantly larger than in the LC condition. Therefore, our findings suggest that the positive-RPE signal represented by dopamine neurons is increased by the cost previously incurred.

The dopamine neurons also showed phasic suppression to the R− cues (Fig. 4d, e). However, the responses of the dopamine neurons to the R− cues did not show a significant difference as a function of cost incurred (two-tailed Wilcoxon's signed-rank test; $P = 0.25$, $n = 70$), and the ROC analysis did not reveal any evidence for a bias in the response distribution (Fig. 4f; Wilcoxon's signed-rank test, $P = 0.35$; $n = 70$). Thus, the paid cost was not reflected in the negative-RPE signal elicited by the nonreward-predicting cues. This may be caused by a floor effect: the spontaneous activity of the dopamine neuron is low (around 5 Hz); and consequently there may not be a sufficient dynamic range to adequately encode any such difference in cost expended for the negative RPE response (Fig. 4d, e).

We also examined the effect of the paid cost on the reward cues for the value and salience type dopamine neurons separately, but both types of dopamine neurons showed similar response pattern (Supplementary Fig. 6a–h). Therefore, the paid cost manifests a

similar effect on the response to reward cues in both the value and salience type dopamine neurons.

The monkeys' actual fixation durations were not constant but varied on a trial-by-trial basis (Fig. 1b). Therefore, it was possible that dopamine responses to the reward cues are modulated by the actual fixation durations on a trial-by-trial basis. However, we could not find any significant correlation between them for each cost and reward condition (Supplementary Fig. 7a–d). Furthermore, RTs to the reward cue were also modulated by the cost and reward conditions (Fig. 1f). One possibility is that the responses of the dopamine neurons could be explained by the RTs to the reward cues on a trial-by-trial basis. However, we could not find any significant correlation between RTs and the normalized dopamine responses to the reward cues (Supplementary Fig. 7e–h). These results suggest that dopamine responses are independent of both RTs and fixation durations on each trial, yet modulated by the amount of required cost and expected reward which are fixed for each type of trials.

Furthermore, it is also possible that the forced aborts in the LC condition generated both the monkeys' preferences and the enhanced activation of the dopamine neurons to the reward cue in the HC condition. If so, the number of forced aborts should be related to both preference and the degree of enhanced activation. However, the number of the forced aborts had no effects on either the monkeys' preference or the activation of the dopamine neurons to the reward cues (Supplementary Fig. 8). Therefore, faster RTs and higher DA responses to the $R_{HC}+$ cue than the $R_{LC}+$ cue are not due to the inserted forced aborts in the LC condition.

**Incurred cost increases dopamine responses to reward delivery**. The response of dopamine neurons to the R+ cues should originate from the response to the reward itself, because dopamine neurons alter their response to reward-predicting cues relative to the stimulus-reward association[8,30]. Therefore, we expected that dopamine neurons would show a paid cost dependent response enhancement to reward delivery. To measure dopamine neuron activity to the reward delivery, the monkeys performed the HLC uncertain task with two novel reward cues (Fig. 5a). Because the rewards were delivered in only half of the reward cue presentations, the reward cues did not reliably nor differentially predict reward delivery. This was done to maximize dopamine neuron responsiveness to the receipt of an (unpredicted) reward, so as to increase our sensitivity to detect a modulation in the responsiveness of the neurons as a function of the cost expended.

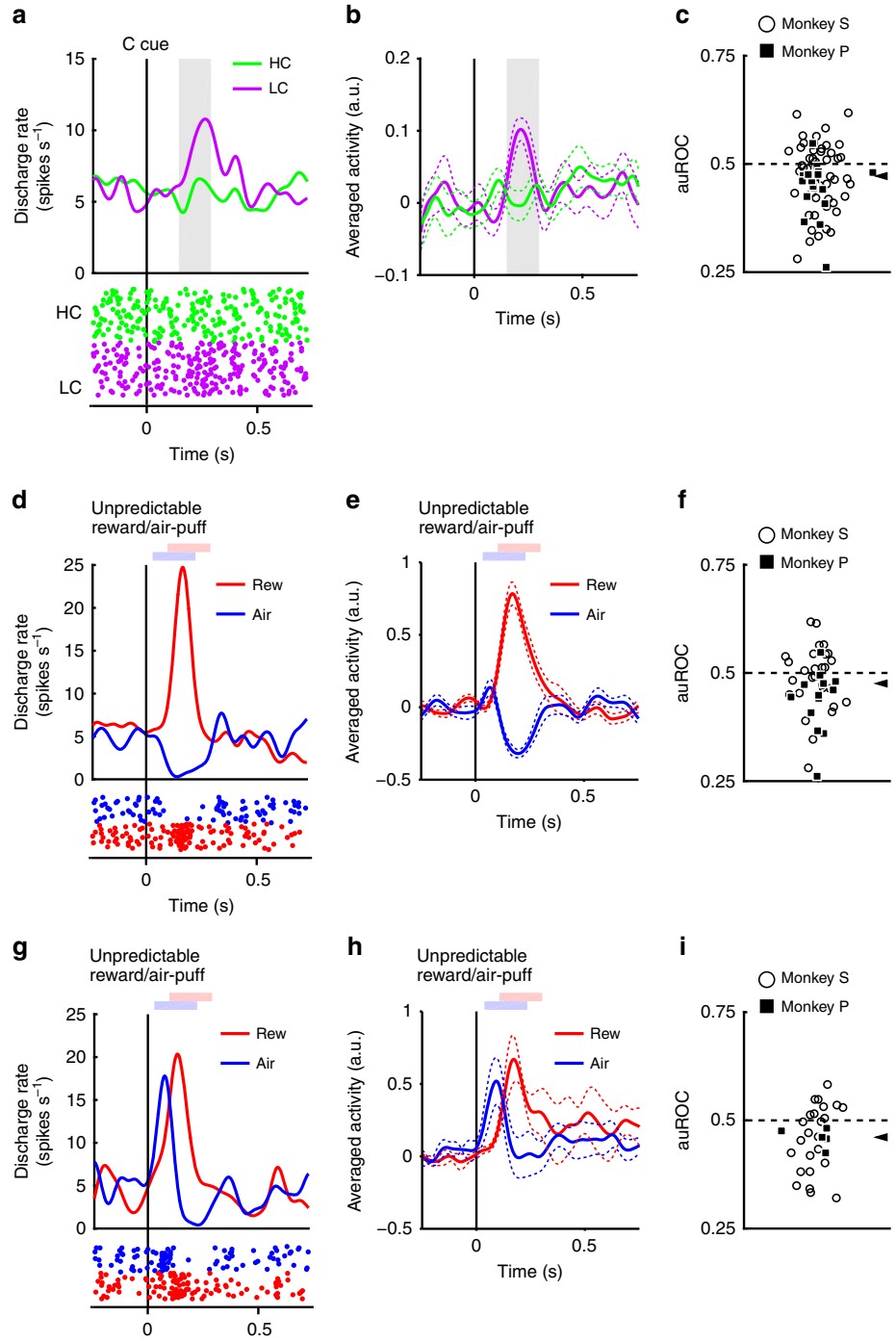

**Fig. 3** Dopamine neuron responses to cost cues. **a** A representative dopamine neuron response to the cost cues. The spike density functions were calculated from the normalized activity of a dopamine neuron recorded from the monkey P. The colored lines indicate the spike density functions and colored dots indicate the spike timing. Green and purple colors indicate activity in the high-cost and low-cost trials, respectively. The vertical line indicates the timing of the cost cue presentation. The gray-colored area indicates the period to calculate the firing rate as the response to the condition cues. **b** Population-averaged activity of the dopamine neurons recorded from monkey P to the condition cues. The solid lines the dashed lines represent mean and SEM, respectively. **c** The distribution of the areas under the ROCs to quantify the effect of the predicted cost on the neuronal response to the cost cue. Filled squares and open circles indicate data from monkey P and S, respectively. The arrowhead indicates the median of the auROC (0.47). **d**, **g** Representative responses of the motivational value type dopamine neuron (**d**) or salience type dopamine neuron (**g**) to the unpredictable reward or air-puff. Red and blue curves indicate the response to unpredictable reward and unpredictable air-puff, respectively. The vertical line indicates the timing of the unpredictable reward or air-puff delivery. Pale red and blue squares indicate the period to calculate the firing rate as the response to the unpredictable reward or air-puff. **e**, **h** Population-averaged activity of the motivational value type dopamine neurons (**e**) or salience type dopamine neurons (**h**) to the unpredictable reward or air-puff. **f**, **i** The distribution of the auROCs calculated from the motivational value type dopamine neurons (**f**) or salience type dopamine neurons (**i**). The arrowheads indicate the medians of the auROCs (**f** 0.48; **i** 0.46)

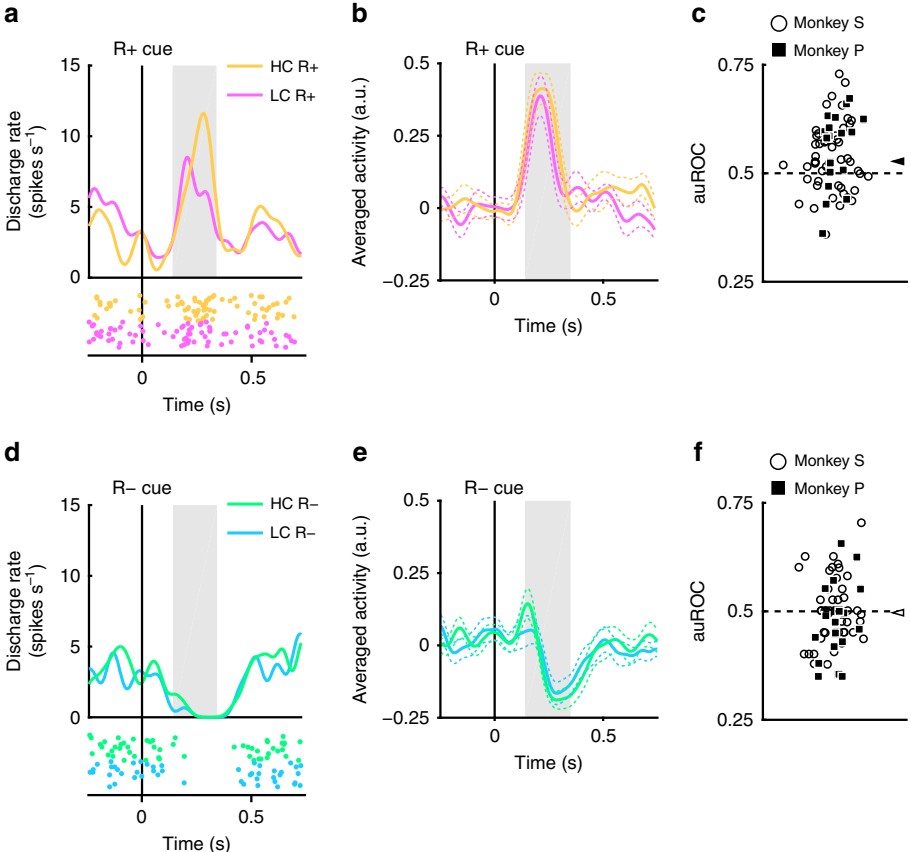

**Fig. 4** Dopamine neuron responses to reward cues. **a** An example neuron response to the R+ cues. The spike density functions were calculated from the activity of dopamine neuron recorded from monkey P. The colored lines and dots indicate spike density and spike timing, respectively. Yellow and pink colors indicate activity in the high-cost and low-cost trials, respectively. The vertical line indicates the timing of the R+ cue presentation. The gray-colored area indicates the period to calculate the firing rate as the response to the reward cues. **b** Population-averaged activity of the dopamine neurons recorded from monkey P to the R+ cues. The solid lines and the dashed lines represent mean and SEM, respectively. **c** The distribution of the auROCs to quantify the effect of the paid cost on the neuronal response to the R+ cues. Filled squares and open circles indicate data from monkey P and S, respectively. The arrowhead indicates the median of the auROCs (0.53). **d** A representative response to the R− cues. Green and cyan colors indicate high-cost and low-cost trials, respectively. The vertical line indicates the timing of the R− cue presentation. **e** Population-averaged activity of the dopamine neurons recorded from monkey P to the R− cues. **f** The distribution of the auROCs to quantify the effect of the paid cost on the neuronal response to the R− cues. The arrowhead indicates the median of the auROC (0.50)

When RTs were compared between the cost cues, monkey P showed a faster RT to the LC cue than the HC cue (Fig. 5b). There was no difference in RTs to the reward cues between the HC and LC condition in either monkey (Fig. 5c).

In the HLC uncertain task, dopamine neurons showed modest activation to the LC cue but did not respond to the reward cues because they were not reward predictive (Fig. 6a). Across the population, evoked responses were smaller to the HC than LC cue (Fig. 6b; two-tailed Wilcoxon's signed-rank test; $P = 2.7 \times 10^{-3}$, $n = 19$), and ROC analyses showed smaller responses to the HC cue (Fig. 6c; two-tailed Wilcoxon's signed-rank test; $P = 5.5 \times 10^{-3}$, $n = 19$). The neuronal response to reward delivery in the HC condition was larger than the LC (Fig. 6d; two-tailed Wilcoxon's signed-rank test; $P = 0.036$, $n = 19$). The distribution of auROCs was >0.5 indicating a larger reward delivery response in the HC relative to LC trials (Fig. 6e; two-tailed Wilcoxon's signed-rank test; $P = 0.049$, $n = 19$). These results indicate that the reward delivery response is enhanced in the HC trial and that paid cost increases the positive-RPE signal at reward delivery.

In addition, we compared dopamine responses following the absence of a reward. The auROCs did not show a biased distribution, indicating that paid cost had no effect on negative RPEs at the time of outcome (Supplementary Fig. 9a). The

dopamine neurons showed no difference between responses to the $R_{HC}$ and $R_{LC}$ cues (Supplementary Fig. 9b).

**Incurred cost enhances learning speed.** Given that RPEs to reward delivery are increased by the paid cost, under the hypothesis that RPEs are directly involved in mediating stimulus-reward learning, we expected enhanced RPEs to be reflected in learning behavior via an enhanced learning speed[24]. To test for an effect of paid cost on learning, the monkeys performed the HLC exploration task (Fig. 7a; see Methods). In this task, two reward cues (R+ and R−) were presented simultaneously and the monkeys had to choose one. We equalized success rates and reward probability between trial types (two-tailed paired $t$ test; $t_{48} = 0.15$, $P = 0.89$, $n = 49$ for monkey P; $t_{85} = 1.2$, $P = 0.25$, $n = 86$ for monkey S). When RTs were compared for the cost cues, both monkeys showed faster RTs to the LC cue than the HC cue (Fig. 7b; two-tailed paired $t$ test; $t_{48} = 12.9$, $P \approx 0$, $n = 49$ for monkey P; $t_{85} = 3.4$, $P = 9.4 \times 10^{-4}$, $n = 86$ for monkey S). When comparing RTs to the reward cues, monkey S showed faster RTs in the HC than LC condition (Fig. 7c; two-tailed paired $t$ test; $t_{48} = 1.3$, $P = 0.19$, $n = 49$ for monkey P; $t_{85} = 2.8$, $P = 6.8 \times 10^{-3}$, $n = 86$ for monkey S). When comparing RTs during the first and latter half of the learning session separately, RTs to the LC cue

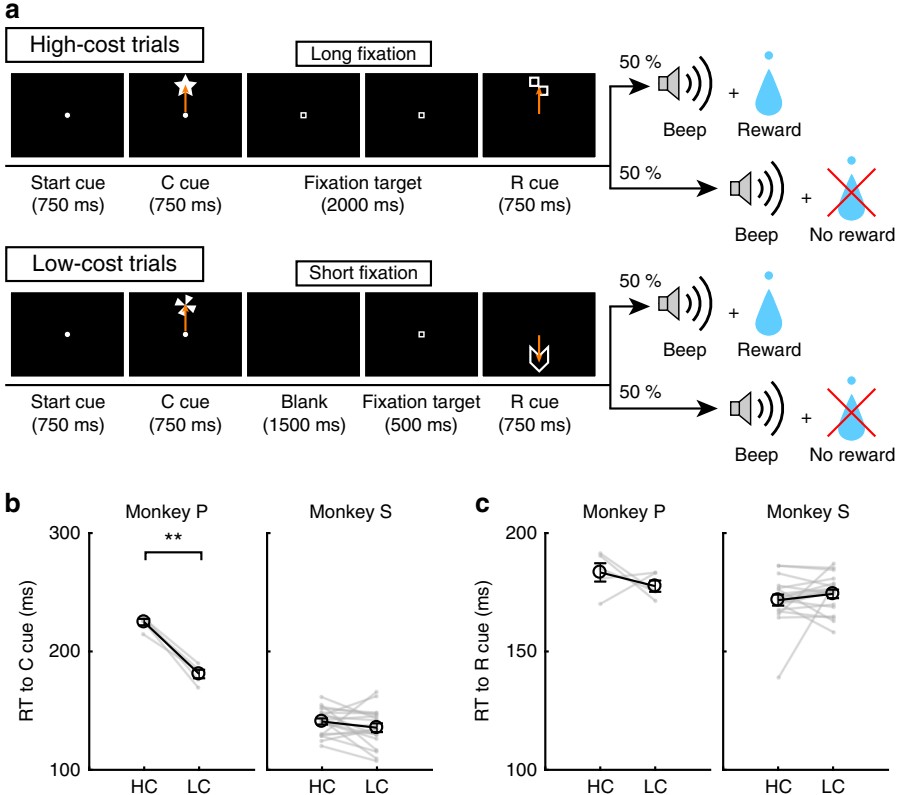

**Fig. 5** HLC uncertain task. **a** The HLC uncertain task. In this task, uncertain reward cues, in which rewards were delivered 50% of the time irrespective of which cue was presented, were used. **b** RTs to the cost cues in the high-cost and low-cost trials. Only monkey P showed a faster RT to the LC cue than the HC cue (**$P < 0.01$; two-tailed paired $t$ test; $t_4 = 9.0$, $P = 8.5 \times 10^{-4}$, $n = 5$ for monkey P; $t_{18} = 1.4$, $P = 0.19$, $n = 19$ for monkey S). Black circles and error bars indicate mean and SEM. **c** RTs to the reward cues in the high-cost and low-cost trials. There was no difference in the RTs to the reward cues between the high-cost and low-cost condition (two-tailed paired $t$ test; $t_4 = 0.97$, $P = 0.39$, $n = 5$ for monkey P; $t_{18} = 0.99$, $P = 0.39$, $n = 19$ for monkey S)

were faster than to the HC cue during the first (Supplementary Fig. 10a) and latter half of the session (Supplementary Fig. 10c). On the contrary, RTs of monkey S to the reward cue in the HC condition were faster than in the LC condition during only the latter half of the session (Supplementary Fig. 10d) but not the first half (Supplementary Fig. 10b).

In the HLC exploration task, reward cues were randomly generated in each learning session. Therefore, the monkeys had to learn the relationship between reward cues and rewards in each session. As trials progressed within a session, the monkeys chose R+ cues more frequently in each cost condition (Fig. 8a). To quantify learning speed, we fit a cumulative exponential function to the data, incorporating two free parameters, $a$ and $b$, indicating the steepness of the curve and the plateau, respectively (Supplementary Fig. 11a, b). The log ratio between steepness parameters (log $a_{HC}/a_{LC}$) was significantly larger than zero indicating a larger steepness parameter in HC than LC trials (Fig. 8b; two-tailed $t$ test; $t_{48} = 2.1$, $P = 0.042$, mean = 0.58, $n = 49$ for monkey P; $t_{85} = 2.5$, $P = 0.013$, mean = 0.19, $n = 86$ for monkey S). The log ratio between plateau parameters (log $b_{HC}/b_{LC}$), was not different from zero indicating no difference between cost conditions (Fig. 8c; two-tailed $t$ test; $t_{48} = 0.76$, $P = 0.45$, mean = −0.0024, $n = 49$ for monkey P; $t_{85} = 0.56$, $P = 0.58$, mean = 0.010, $n = 86$ for monkey S). These results indicate that learning speed is faster in the HC trials. Next, we modeled learning curves using a reinforcement-learning (RL) model (see Methods). This model includes learning rate parameters ($\alpha_{HC}$ and $\alpha_{LC}$) and exploration rates ($\beta_{HC}$ and $\beta_{LC}$) for both cost conditions (Supplementary Fig. 11c, d). When fitting to behavior, we found that the log ratio between learning rate parameters (log $\alpha_{HC}/\alpha_{LC}$)

was larger than zero indicating a significantly larger learning rate parameter in HC than LC trials (Fig. 8d; two-tailed $t$ test; $t_{48} = 2.3$, $P = 0.026$, mean = 0.50, $n = 49$ for monkey P; $t_{85} = 2.2$, $P = 0.034$, mean = 0.25, $n = 86$ for monkey S) while the parameter $\beta$ showed no difference (Fig. 8e; two-tailed $t$ test; $t_{48} = 0.77$, $P = 0.44$, mean = 0.0097, $n = 49$ for monkey P; $t_{85} = 0.64$, $P = 0.52$, mean = 0.038, $n = 86$ for monkey S). Here, we estimated the learning rate parameters for each cost condition ($\alpha_{HC}$ and $\alpha_{LC}$) separately to explain faster learning speeds in the HC condition. However, if learning rates are identical between the cost conditions, the ratio between the estimated learning rate parameters ($\alpha_{HC}/\alpha_{LC}$) can be thought of as an amplification value for RPEs in the HC condition. Therefore, these results suggest that an amplification of RPEs can explain faster learning speeds in the HC condition.

We also tried to explain the learning process with alternative RL models which take into account a possibility that the monkeys know the anticorrelation between stimuli and reward on each trial. In those models the value of the unchosen option is updated alongside the chosen one (Supplementary Fig. 12). Even when applying such alternative models to the data, the learning rate parameter was significantly larger in the HC compared to the LC condition (Supplementary Fig. 12b, f) while the parameter $\beta$ showed no difference (Supplementary Fig. 12d, h). Thus, our finding about an amplification of the RPE signal in the HC condition is robust to the form of RL model fit to the data.

## Discussion

We investigated the effect of paid cost on the value of reward-predicting cues and on the phasic responses of midbrain

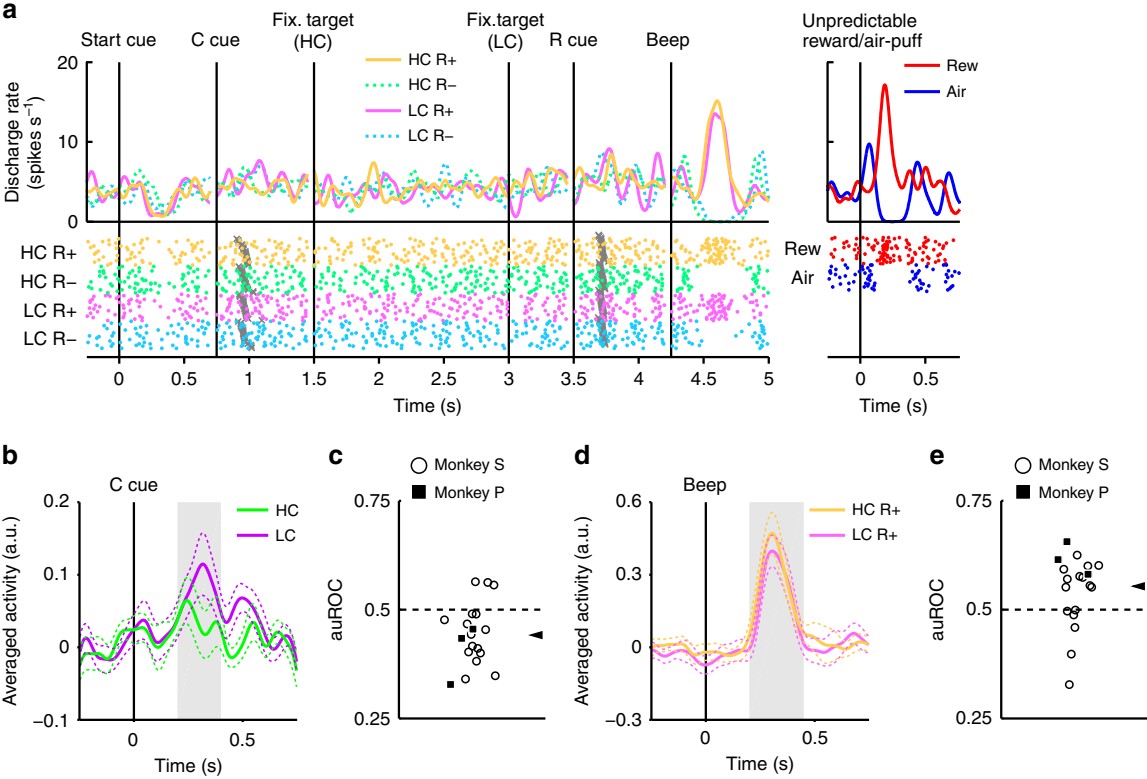

**Fig. 6** Dopamine neuron response to the reward delivery. **a** Representative dopamine neuron activity in the HLC uncertain task. Each color represents the conditions (yellow: HC+, green: HC−, pink: LC+, cyan: LC−). The timings of the saccade onset are indicated by gray crosses. The responses of this dopamine neuron to the unpredictable reward and air-puff are also depicted in the right panel (red: unpredictable reward, blue: unpredictable air-puff). **b** Population-averaged activity of the dopamine neurons recorded from monkey S to the condition cues. Green and purple colors indicate activity in the high-cost and low-cost trials, respectively. The solid lines and the dashed lines represent mean and SEM, respectively. Gray-colored area indicates the time window to calculate the firing rate as the response to the condition cues. **c** The distribution of the auROCs to quantify the effect of the predicted cost on the neuronal response to the cost cues. Filled squares indicate the data from the monkey P ($n = 3$) and open circles indicate the data from the monkey S ($n = 16$). The arrowhead indicates the median of the auROC (0.44). **d** Population-averaged activity of the dopamine neurons recorded from the monkey S to the reward delivery. Yellow and pink colors indicate activity in the high-cost and low-cost trials, respectively. Gray-colored area indicates the time window to calculate the firing rate as the response to the reward delivery. **e** The distribution of the auROCs to quantify the effect of the paid cost on the neuronal response to the reward delivery. The arrowhead indicates the median of the auROC (0.55)

dopamine neurons. Monkeys showed increased valuation for reward-predicting cues following the performance of an action that incurred a larger cost. Dopamine neurons showed increased responses to both the reward-predicting cue and reward delivery, after a higher cost had been incurred. Furthermore, the monkeys showed faster learning speeds when a higher cost was required to obtain reward.

Several studies have shown that a paid cost enhances preferences for a reward-predicting cue[1–3]. In the present study, the monkeys showed faster RTs to the reward-predicting cues in the HC condition compared to those in the LC condition, consistent with the possibility that the reward cue value is enhanced by the paid cost[27]. An alternative possibility is that the longer fixation time associated with enhanced attention to the saccade target in the HC condition, therefore, reducing RTs after a longer fixation in the HC trial. However, we did not find any difference between RTs to the R cues in the HC and LC trials in the HLC uncertain task. Furthermore, in the first half of the HLC exploration session. RTs to the R cues were not significantly different between the HC and LC trials. These findings thus indicate that a longer fixation is not a likely explanation for the shorter RTs observed to the reward-predicting cues. In addition to the effect of paid cost on reward cue RTs, the cost also affected RTs to the nonreward-predicting cues, despite the fact that no reward was delivered after the cue presentations. A previous study reported a similar

phenomenon, in that monkey subjects showed shorter RTs in unrewarded trials when more preferred rewards were employed in the alternate trials within each block[30]. One possible interpretation of the effect in that study is that an overall higher motivation to respond in the block with more preferred reward affected RTs also to the no reward cue within the block. Similarly, in the present study, the expectation of a more valuable reward in the HC trials might have modulated RTs to the no reward cue in the HC trials in our task. Furthermore, the effect of the paid cost on RTs to the reward cues was smaller than that to the no reward-predicting cues. This is likely an artifact of the fact that because the monkeys made a saccade more rapidly to the $R_{LC}+$ cue in the first place, there is reduced scope to detect a shortening of RTs to the $R_{HC}+$ cue. Therefore, the difference in RTs between the R+ cues would be small as a consequence.

The monkeys also performed choice trials between the $R_{HC}$ and $R_{LC}$ cues in the HLC task. However, while monkey S showed a preference for the $R_{HC}+$ cue to the $R_{LC}+$ cue, monkey P showed no such preference. This discrepancy could be explained by a contextual difference between the HLC saccade and choice trials. In the choice trials, two reward-predicting cues were presented instead of one reward-predicting cue. Furthermore, the monkeys obtained no reward after their choice even if they chose the reward-predicting cue, thus the choice test was done in extinction. The extinction procedure was implemented to ensure

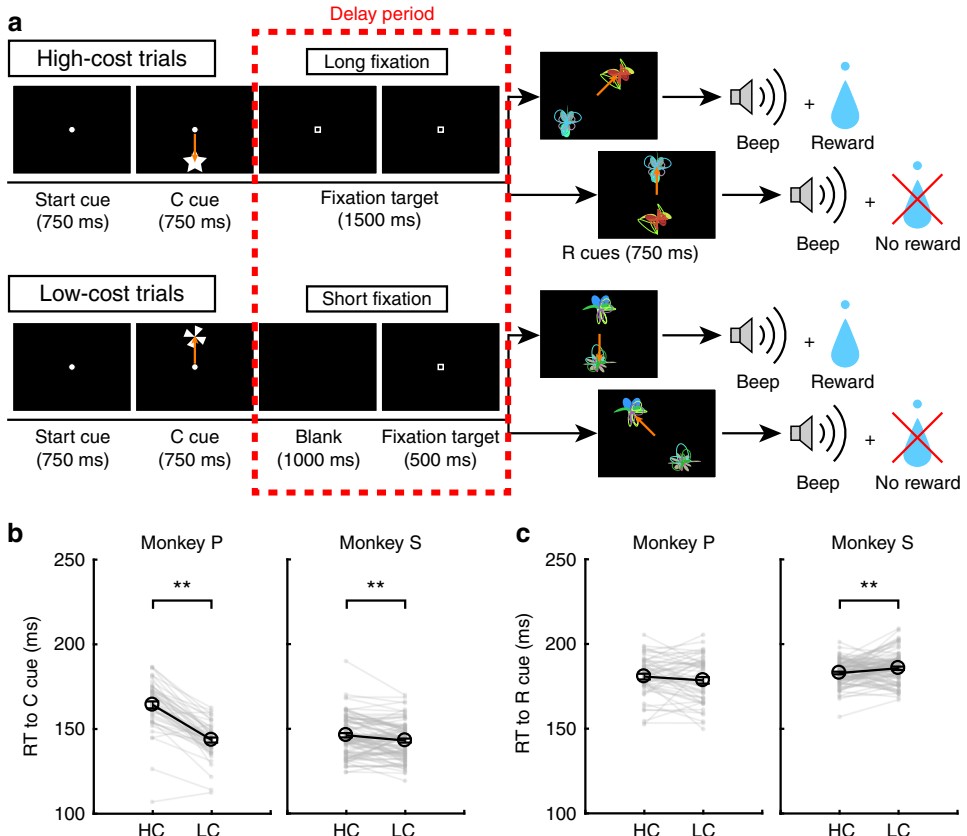

**Fig. 7** HLC exploration task. **a** The HLC exploration task. In this task, monkeys had to choose between R+ and R− cues, which were randomly generated in each learning session. If they chose the R+ cue they could obtain a reward and if they chose R− cue they would not obtain reward. **b** RTs to the cost cues in the high-cost and low-cost trials. The monkeys showed faster RTs to the low-cost cue (**P < 0.01; two-tailed paired t test). Black circles and error bars indicate mean and SEM. **c** RTs to the reward cues in the high-cost and low-cost trials. Monkey S showed faster RTs to the reward cues in the high-cost condition

the monkey's choice was driven by what had been learned on the effort trials, as opposed to being confounded with new learning on the choice trials. However, this procedure may have the side effect that the monkey could quickly learn to recognize the extinction procedure in the choice context and that there is no reason to choose the more preferred stimuli. Nevertheless, one of the monkeys did in fact show a preference for the reward cue in the HC condition.

At the time of presentation of a cue that predicted a subsequent requirement to pay a cost, the activity of dopamine neurons was reduced, consistent with previous studies[22,23]. In our study, we did not observe an overall decrease in dopamine neuron responding to both HC and LC cues relative to baseline. This suggests that a negative-RPE signal does not occur at that time-point in spite of the following cost. The absence of negative RPE presumably reflects integration of a prediction of future reward expected later in the trial. The dopamine neurons showed significant activation in the LC trial and the activity was higher compared to the HC trial. This suggests that cost information is incorporated into the RPE signal carried by dopamine neurons. Thus, dopamine neurons code both reward and cost information and the RPE response reflects the sum of cost and reward.

We demonstrated that the RPE signal represented by dopamine neurons is enhanced by the paid cost at the point of reward cue presentation (in the HLC saccade task) and reward delivery (in the HLC uncertain task). The objective amount of reward delivered in the HC and LC trials was equal; therefore, modification of the RPE signals should be caused by a nonsensory process. This possibility is supported by several studies indicating

a contextual effect on dopamine RPE signals consistent with the processing of the subjective value and/or utility in dopamine neurons[11,19–21,31–34]. If the RPE signal is larger, this should produce a more rapid updating of the cue value, which would consequently impact on the learning speed of stimulus-reward associations. Previous studies have shown a modification of learning speed by nonsensory factors[24,35]. In line with this, the monkeys exhibited faster learning speeds in the HC relative to the LC condition. We found that enhanced learning speed by the paid cost can be explained by an RL model with an amplified RPE. It was difficult to separate the effects of the amplified RPE and increased learning rate in our experiments; however, we found an amplified dopaminergic RPE signal in the HC condition. Furthermore, a prior fMRI study has shown that the learning rate parameter is represented in the anterior cingulate cortex and that activity of the VTA is not related to the learning rate parameter in volatile environments[36]. Therefore, we argue that the RPE signal coded by dopamine neurons is amplified by the paid cost, and that the increased RPE signal enhances the learning speed.

When the RPE signal was generated at the time of the reward cue presentation and the reward delivery, the monkeys had already paid the cost. Therefore, one possible mechanism for the enhanced RPE signal is that a reward obtained after a HC might be more rewarding. An increased expectation of a more valuable reward after the HC might enhance the motivation to finish the trial, thereby shortening the RT to the reward cues in the HC trials.

Another possible interpretation of our results is that relief experienced from the termination of the costly action may act as a

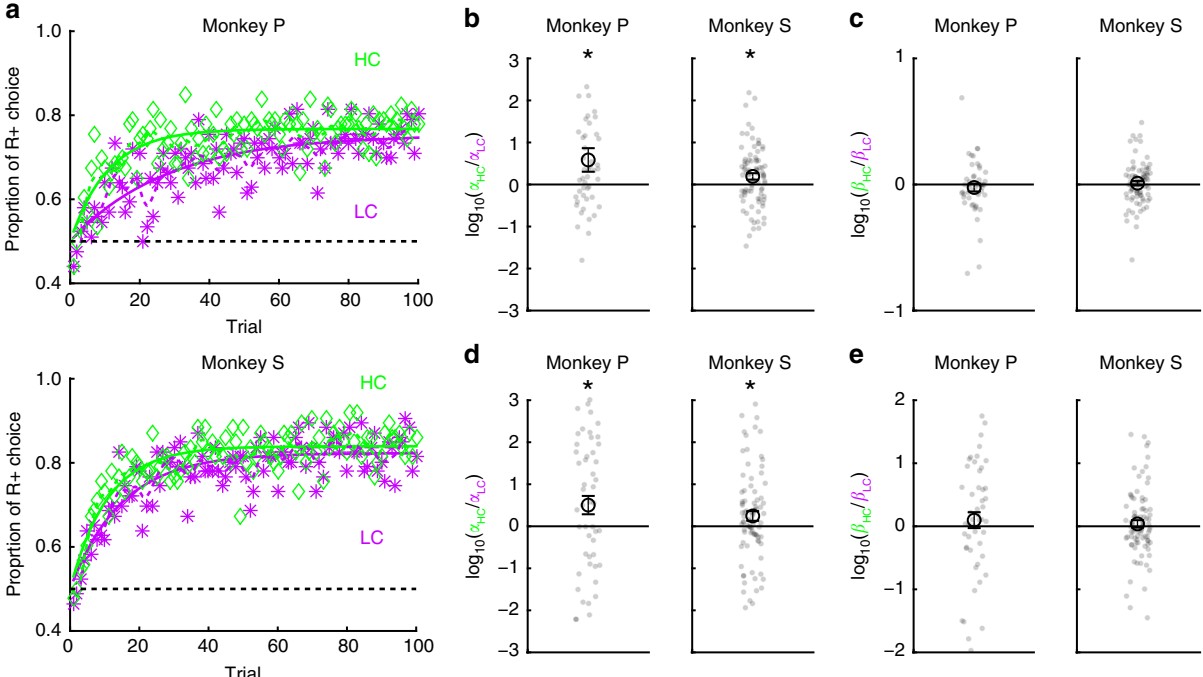

**Fig. 8** Learning speed test. **a** Mean learning process of monkeys P and S. The proportion of R+ choices is plotted as a function of the trial. The green and purple points indicate data from high-cost and low-cost trial, respectively. The dotted lines represent smoothed learning process. The cumulative exponential functions were fitted to the data points and represented as the solid lines. **b** The log ratio between the fitting parameters $a$ in the high- and low-cost conditions when the data were fit with a cumulative exponential function (*$P < 0.05$; two-tailed Wilcoxon's signed-rank test). Black circles and error bars indicate mean and SEM. **c** The log ratio between the fitting parameters $b$ in the high- and low-cost conditions when the data were fit with a cumulative exponential function. **d** The log ratio between the learning rate parameter $\alpha$ in the high- and low-cost conditions when the data were fit with a reinforcement-learning model. **e** The log ratio between the fitting parameter $\beta$ in the high- and low-cost conditions when the data were fit with a reinforcement-learning model

reward for the monkeys. Functional magnetic resonance imaging (fMRI) studies have shown that pain relief may be a reward for human participants[37,38]; therefore, cost might play a similar role as an aversive stimulus to pain. If the relief from cost is rewarding and if this is reflected in dopaminergic activity, we would expect that the dopamine neurons would respond at the end of the long fixation, which is the timing of reward cue presentation. However, we did not observe any difference in dopaminergic activity at the timing of reward cue presentations nor any difference in RTs to the R cues between the HC and LC trials in the HLC uncertain task. Therefore, we suggest that relief from cost is not providing an adequate explanation for the effect we observed in the dopamine neurons.

Furthermore, the dopamine neurons showed qualitatively different responses to the aversive stimulus compared to the cost predicting cue. One possible explanation for this is that the effort cost was less salient than the air-puff or reward, because the effort cost was temporally extended across several seconds as the monkeys performed the fixation and not punctate. Therefore, dopamine neurons may not have been activated to the less salient cost cues. Another possibility is that the salience type dopamine neurons respond to events after which some movements were induced. When the reward or the air-puff was delivered to the monkeys, they make some movements such as the licking or the eye blink. However, in the HLC saccade task, the monkeys had to keep their gaze on the fixation target without any movement as the cost. Actually, a recent study showed that dopamine release in the nucleus accumbens following a reward-predicting cue is attenuated unless movement is correctly initiated[39]. Because the cost in our experiments did not involve movement initiation, this might potentially result in an inconsistent response of salience

type dopamine neurons. Either way, we can conclude that cost information is processed differently from aversive information.

In conclusion, we suggest that paid cost increases the value of reward-predicting cues and that this in turn increases the RPE signal coded in the midbrain dopamine neurons. This effect led to a behavioral prediction that an animals' rate of learning would be enhanced for reward-predicting cues following the experience of a HC. This is indeed what we observed. Thus, our observations on the activity of dopamine neurons led us to hypothesize the existence of a behavioral effect, as well as a putative computational mechanism underlying this effect, which we subsequently confirmed. Our findings therefore represent an example of how triangulation can happen between measurements of neural data, computational theory and behavior: developing a deeper understanding of neuronal processing in the brain can yield insights about behavior and its underlying computational basis.

## Methods

**Animals**. We used two male Japanese monkeys (*Macaca fuscata*; body weight, 6.5 kg = Monkey P; body weight, 9.0 kg = Monkey S). We implanted a head post on the top of the monkey's skull so that it could be fastened to a chair at a later date. A recording chamber was also implanted to enable the mounting of an electrode micromanipulator. The recording chamber was tilted 45° laterally in the coronal plane and positioned at the stereotaxic coordinates: 15 mm anterior to the external canal. After a recovery period, the monkeys were trained to perform the saccade task. After completing the training, we drilled a hole through the skull inside the recording chamber for electrode insertion. All animal care protocols were approved by the Animal Experiment Committee of Tamagawa University, and conformed to the National Institutes of Health Guide for the Care and Use of Laboratory Animals.

**Behavioral task**. Monkeys were trained to perform the HLC saccade task (Fig. 1a), HLC uncertain task (Fig. 5a), and HLC exploration task (Fig. 7a). All tasks were

performed in a dark room. The monkeys were seated in a chair in front of a 22-in. LCD monitor (S2232W, Eizo) with their implanted head posts fixed to the chair. The distance between their eyes and the display was 70 cm. When a start cue (white circle, 0.3° diameter) was presented at the center of the display, the monkey was required to maintain its gaze on the cue. The start cue disappeared after 750 ms and then a cost cue was presented (star and windmill for the HC and LC trials, respectively). Monkeys were required to saccade to the cost cue during the 750 ms of cue presentation. If they did not saccade to the cue, the trial was aborted and the same trial started again. During HC trials, the fixation target (0.3° × 0.3° white square) was presented just after the disappearance of the cost cue for 2000 ms (HLC saccade and HLC uncertain tasks) or 1500 ms (HLC exploration task) and the monkeys were required to saccade to it and keep their gazes on it. If the monkeys moved their gaze beyond a fixation window of 4° × 4°, the task was aborted. The fixation window was activated 400 ms after the fixation point presentation because the monkeys needed time to prepare for the saccade and for adjustment of their fixation. Therefore, the monkeys had to fixate for at least 1600 ms (HLC saccade and HLC uncertain tasks) or 1100 ms (HLC exploration task) in the HC trials. In the LC trials, a blank screen was displayed for 1500 ms (HLC saccade and HLC uncertain tasks) or 1000 ms (HLC exploration task), and then the fixation target appeared for 500 ms. Because the fixation window was activated 400 ms after fixation point presentation, the monkeys were required to fixate on the target for at least 100 ms in the LC trials. The monkeys exhibited more errors in the HC trial; therefore, a forced abort was randomly inserted 100 ms before the reward cue presentation (400 ms after the fixation target presentation which is the timing of fixation window onset) in the LC trial to equalize the success rate. After fixating on the target, one or two reward cues were presented and the monkeys were required to saccade to the cue. If they successfully made a saccade to the reward cue, a beep sound was delivered 750 ms after the reward cue presentation. When the monkeys made a saccade to the R+ cue, 0.3 ml of water was delivered at the same time as the beep. No reward was delivered when they made a saccade to the R− cue.

In the HLC saccade task, four colored circles were used as reward cues (R$_{HC}$+: yellow; R$_{HC}$−: green; R$_{LC}$+: pink; R$_{LC}$−: blue; Fig. 1a). One experimental session consisted of 80 saccade trials, 20 unpredictable reward trials, 20 unpredictable air-puff trials, and 5 choice trials. The saccade trials, included 40 HC trials and 40 LC trials, both of which included 20 reward trials and 20 no reward trials. In the unpredictable reward or air-puff trials, 0.3 ml of water reward or air-puff (150 ms for monkey P; 200 ms for monkey S) was delivered to the monkeys' face without being cued. The choice trials included a trial in which monkeys made a choice between R+ cues (R$_{HC}$+ vs. R$_{LC}$+) in the HC trial, between R− cues (R$_{HC}$− vs. R$_{LC}$−) in the HC trial, between R+ (R$_{HC}$+ vs. R$_{LC}$+) cues in LC trial, between R− (R$_{HC}$− vs. R$_{LC}$−) cues in LC trial, and between the cost cues (Supplementary Fig. 1). In trials with a choice between reward cues, the task structure was identical to the saccade task before reward cue presentation. Next, instead of presenting a reward cue, two reward cues were presented in the choice trials and no reward was delivered after reward cue presentation even if the monkeys made the choice between R+ cues.

To test the response of dopamine neurons to reward delivery, monkeys performed the HLC uncertain task (Fig. 5a). This task was similar to the HLC saccade task except for the reward cues. In this task, we used two reward cues (instead of the four reward cues used in the HLC saccade task), one for the HC trial and the other for the LC trial. The reward was delivered in half of the trials after reward cue presentation. One experimental session consisted of 80 saccade trials, 20 unpredictable reward trials, and 20 unpredictable air-puff trials. The saccade trials included 40 HC trials and 40 LC trials, both of which included 20 reward trials and 20 no reward trials. In the unpredictable trials, a reward or air-puff was delivered without any cue.

In the HLC exploration task, two reward cues (R$_{HC}$+, R$_{HC}$− or R$_{LC}$+, R$_{LC}$−) were presented simultaneously and the monkeys were required to saccade to one of the reward cues (Fig. 7a). If they chose the R+ cue, they were provided with a water reward. Four reward cues (R$_{HC}$+, R$_{HC}$−, R$_{LC}$+, R$_{LC}$−) were generated for each exploration session and the monkeys were required to learn the association between the cues and reward trial-by-trial. One experimental session consisted of 100 HC trials and 100 LC trials. We found that for the exploration task, if we set the fixation duration to be 2000 ms in the HC condition so that it matched the duration of HC condition in the other tasks, the monkeys performed the task with a very low success rate perhaps because of the difficulty of the task and/or the consequent low reward rate. Therefore, to reduce the difficulties of the task and increase the success rate, we used a 1500 ms fixation duration as the cost for the HLC exploration task.

The tasks were controlled using a commercially available software package (TEMPO, Reflective Computing, St. Louis, MO, USA). A custom-made program using an application programming interface (OpenGL) was used for visual stimulus presentation. The visual stimuli for the cost and the reward cues were created by the authors.

**Recording and data acquisition.** The location of the substantia nigra was estimated using MR images. An epoxy-coated tungsten electrode (shank diameter, 0.25 mm, 0.5–1.5 MΩ measured at 1000 Hz, FHC) was inserted into the substantia nigra using a micromanipulator (MO-972, Narishige, Tokyo, Japan) mounted onto

the recording chamber with a stainless guide tube. Voltage signals were amplified (×10,000) and filtered (0.5–2 kHz). Action potentials from a single neuron were isolated with a template-matching algorithm (OmniPlex, Plexon, Dallas, TX, USA). Eye movement was monitored by an infrared camera system at a sampling rate of 500 Hz (iView X Hi-Speed Primate, SMI, Teltow, Germany). The timing of action potentials and behavioral events were recorded with a time resolution of 1 kHz.

**Data analysis.** To analyze the monkeys' behavior, RTs were determined as the time interval between stimulus onset and the time when monkeys initiated the saccade. The saccade initiation was determined by calculating the timing when the gaze position exceeded 5 standard deviations from the mean gaze position prior to cue presentation.

In the HLC exploration task, the monkeys' choice behavior was quantified by fitting a cumulative exponential function. The function (P) describes the proportion of correct choice as follows:

$$P = \frac{1}{2} + \left( \frac{1}{2} - \frac{1}{2} \cdot \exp(-a \cdot t) \right) \cdot b, \tag{1}$$

where $t$ means trial, $a$ and $b$ indicate the slope and plateau of the curve, respectively. This function was fit independently to the data for the two cost conditions. The parameters of the function were searched to maximize the likelihood of observing the data from a single session and the averaged data. A bootstrap method was applied to estimate the confidence intervals when fitting to the averaged data. A standard RL model was also used to quantify the behavioral data. The stimulus values $V_j(t)$ for the selected choice $j$ ($j = 1$ for HC condition; $j = 2$ for LC condition) were updated as follows:

$$V_j(t + 1) = V_j(t) + \alpha_j \cdot \left( R(t) - V_j(t) \right), \tag{2}$$

where $\alpha_j$ indicate the learning rates, which were constrained to values between 0 and 1. $R(t)$ indicates the reward amount (1: rewarded, 0: no reward) at trial $t$.

The probability $P_j(t)$ of choosing stimulus $j$ out of the two stimuli at trial $t$ is given by the softmax rule

$$P_j(t) = \exp\left( \frac{V_j(t)}{\beta_j} \right) \Big/ \sum_{i=1}^{2} \exp\left( \frac{V_i(t)}{\beta_i} \right), \tag{3}$$

where $\beta_j$ indicates the extent of the exploration.

We recorded neuronal activity during the HLC saccade and HLC uncertain task but not the HLC exploration task. The HLC exploration task was implemented as a purely behavioral study. Dopamine neurons were identified if they exhibited each of the following properties: a low tonic firing rate (<6 Hz), a long duration of the spike waveform (>300 μs), and a phasic response to the unpredictable reward (Supplementary Fig. 2a). We analyzed trials in which the monkeys could complete the trial without any errors (braking fixation, no saccade or artificial abort). The mean neuron firing rate was calculated with 1 ms bins and smoothed with a Gaussian kernel ($\sigma = 30$ ms, width $= 4\sigma$) to produce spike density functions. Responses of the dopamine neurons to each task event were calculated as the normalized firing rate relative to the spontaneous activity (mean firing rate during the 500 ms before start cue onset). The firing rates were calculated within time windows determined for each task event and subject. These time windows were determined from the population-averaged activity. We defined the start and the end points of time windows determined based on the rise and fall time of the population-averaged response using previous monkey dopamine studies as references (Supplementary Fig. 3). The time window for the start cue was defined as 200–400 ms after start cue onset for neurons recorded from monkeys P and S. The time window for the condition cue was defined as 150–300 ms for monkey P and 200–400 ms for monkey S. The time window for the reward cue was defined as 140–350 ms after reward cue onset for monkey P and 220–420 ms for monkey S. The time window for the reward delivery was defined as 225–475 ms after the beep onset for monkey P and 200–450 ms for monkey S. The time window for the unpredictable reward delivery was defined as 100–300 ms after the reward delivery for monkey P and 150–300 ms for monkey S. The time window for the unpredictable air-puff was defined as 30–230 ms after the air-puff delivery for monkey P and 50–200 ms for monkey S.

We classified all recorded dopamine neurons into two distinct categories, motivational value and salience types. If the response of a neuron to the air-puff stimuli was smaller than the spontaneous activity, the neuron was classified as being of the motivational value type (Fig. 3d, e). In contrast, if the response of a neuron to the air-puff stimuli was larger than the spontaneous activity, the neuron was classified as being of the salience type (Fig. 3g, h).

To quantify differential neuronal activity between task conditions, an ROC analysis was performed. We calculated the auROC for each neuron. The auROC smaller or larger than 0.5 indicates a smaller or larger response in the HC trial, respectively. Because the numbers of neurons in some neuronal data sets were small, we used Wilcoxon's signed-rank test to reduce the effect of the outliers for quantifying the biased distribution of the auROCs.

Commercially available software, MATLAB (MathWorks, Natick, MA, USA), was used to perform all data analysis.

**Histological examination**. After the recording experiment, both monkeys were euthanized and histological analysis was performed to verify the recording position (Supplementary Fig. 2b). Monkeys were euthanized by administration of a lethal dose of pentobarbital sodium (70 mg kg$^{-1}$) and perfused with 4% formaldehyde in phosphate buffer. Serial coronal sections (thickness, 10 μm) were cut and immunostained with anti-tyrosine hydroxylase (TH) antibody (every 25 sections; anti-TH antibody, 1:500; Merck, Darmstadt, Germany) or Nissl staining (every 25 sections).

**Reporting summary**. Further information on research design is available in the Nature Research Reporting Summary linked to this article.

## Data availability
The data used in the analysis of this study are available from the corresponding author upon reasonable request. A reporting summary for this article is available as a Supplementary Information file. The source data underlying Figs. 1, 3–8 and Supplementary Figs. 1, 4–12 are provided as a Source Data file.

## Code availability
Matlab codes used in the analysis of this study are available from the corresponding author upon reasonable request.

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

## Acknowledgements
This work was supported by MEXT/JSPS Grants-in-Aid for Scientific Research (Kakenhi) Grant numbers JP16H06571 and JP18H03662 to M.S. This research was partially supported by the Strategic Research Program for Brain Sciences supported by Japan Agency for Medical Research and Development (AMED) and the Japan-U.S. Brain Research Cooperation Program. This research was supported by the National Bio-Resource Project at National Institute of Physiological Science (NBRP at NIPS) from Japan Agency for Medical Research and Development, AMED. We thank Bernard W. Balleine and Andrew R. Delamater for their help on writing the paper.

## Author contributions
S.T., J.P.O. and M.S. designed the experiments. S.T. performed the experiments, and analyzed the data. J.P.O. and M.S. refined the experiments and the data analyses. S.T., J.P.O. and M.S. wrote the manuscript.

## Additional information

**Competing interests:** The authors declare no competing interests.

