## [Peer Review File · Nature Communications]

Reviewers' Comments:

Reviewer #1:

Remarks to the Author:

Reward-seeking behavior is fundamental to survival. Several factors such as the value of reward influence this behavior. The cost paid to obtain the reward is also one of the important factors though only a few studies have investigated the neural substrates involved in processing cost information.

The present study tackled the issue about whether and how dopamine neurons contribute to processing cost information. Since the dopamine system attracts much attention as a key structure that regulates reward-seeking behavior, the issue challenged by the present study is timely and would have a strong impact on the field.

In the present study, the authors designed interesting tasks in which monkeys had to pay a cost, i.e., fixation time, to obtain a liquid reward. In those tasks, the monkeys seemed to prefer the reward obtained after paying a high cost compared with a low cost, as we experience in daily life. The authors found that the reward related activity (i.e., the reward prediction error signal) of dopamine neurons was enhanced after the monkey paid a high cost compared with after the animal paid a low cost. More interestingly and surprisingly, the animals learned more quickly in a choice task under a high-cost condition than under a low-cost condition, which was expected from the dopamine data and a reinforcement learning model. These data suggest that dopamine neurons are crucial substrates for processing cost information, and imply that the cost paid to obtain rewards influences reward-seeking behavior through the effect of the cost on dopamine neuron activity.

Their task design is unique, and the findings are new and important to understand the neural mechanism underlying reward-seeking behavior. Before decision, I would like to see responses from the authors to the following my comments.

(1) The monkeys made more errors in high-cost trials than in low-cost trials. In order to compensate the success rate (i.e., reward probability), the authors inserted a forced abort in the low-cost trials. This manipulation made the reward prediction error equal between the two cost conditions, and enabled the authors to analyze dopamine neuron activity related to the cost but not the prediction error. This seems to be a clever way to compensate the prediction error. However, I have a concern; if trials in which monkeys have successfully performed are aborted, the monkey's motivation to accomplish the trials would decrease. Such a decrease in motivation affects (probably decreases) the subjective value of reward obtained in the trials. Thus, although the authors reported that the monkeys preferred the reward after high cost and that dopamine neurons were more strongly activated by the reward after high cost, these effects on the monkey's preference and dopamine activation could be explained by the decrease in motivation in the low-cost trials rather than cost itself. This concern must be solved by new analyses or discussed properly.

(2) In Figure 1e, the authors showed that RT to Rcue is shorter in high-cost trials than in low-cost trials. Based on this data, the authors postulated that the monkeys more preferred the reward in high-cost trials. However, the effect of cost on RT to Rcue is much larger in R- cue than R+ cue. This data seems to conflict with the authors' explanation, because the monkey did not obtain the reward after R-cue. If the monkeys really preferred the reward in high-cost trials, the effect of cost on RT has to be larger for R+ cue.

(3) In the present study, the authors presented the PSTHs of example neurons and the scatter plot for population analyses, but no population PSTH. The population PSTH is more helpful to understand the entire trend of dopamine neuron response, and most of the previous electrophysiological studies on

dopamine neurons have shown the population PSTH. The authors need to present it.

(4) In the present study, the authors reported that dopamine neurons were suppressed by the start cue. This is unusual. Previous studies have reported that dopamine neurons are activated by cues (e.g., fixation point) indicating trial start. Why did dopamine neurons show the suppression?

Reviewer #2:

Remarks to the Author:

In this study, Tanaka, O'Doherty and Sakagami measure activity from putative dopamine neurons in substantia nigra in two macaque monkeys and three different tasks and show that firing rates are modulated not just by reward but also by the cost that monkeys have to pay (keeping a longer vs shorter fixation). While responses are higher for a low-cost than a high-cost cue, once the cost has been paid, the response to a reward cue is enhanced in high-cost trials. Paying a high cost also speeds up learning in a separate reward-learning task.

This is a carefully designed study with an impressive set of sub-experiments which directly provide replications of some of the reported effects. The study asks an important and timely question – whether the dopaminergic system encodes or even integrates costs, in addition to encoding reward prediction errors. The results look promising and could make an important contribution to the field. However, I think in its current form, the results are not always clear and convincing, and some analysis choices are not well-motivated. The analysis and presentation of the data needs to be improved to clarify some of the major points.

(1) The temporal evolution of the encoding of cost and reward information is only shown for representative neurons (and it is not well described how these neurons were chosen). The visualizations of the key results thus rely on a small set of neurons. It would be a lot more convincing if time-course plots could show population average responses: this would provide an intuition about the effect sizes and the peak timing of the effects. While the effects seem quite consistent and robust, the way the results are currently plotted, it is hard to be sure and the plots are not very intuitive. The statistics are done on the auROC of all neurons which rely on set time windows in which the signal is evaluated. The choice of these time windows is somewhat unclear. Were they defined based on the animals RT? If so, how was the start point of each window determined? Could significance be reported in sliding windows rather than for one fixed window? Please justify the choice of Wilcoxon signed-rank tests.

(2) While the learning effects in the third ('explore') experiment seem robust, several pieces of information are missing. The learning rate α is not currently reported, and it is hard to interpret the values of the critical parameter $wRPE$ without knowing α . Is it also unclear if $\alpha * wRPE$ as the product is restricted to be <1 or whether α can vary between 0-1 and $wRPE$ take any values? In the latter case the product (effective learning rate in HC trials) could become larger than 1 which would make it difficult to interpret (and $wRPE$ goes up to 10 in some cases in Fig 8d which it should only do if the learning rate was very low <0.1). Also, was α fitted on all trials? Fig 8a shows that learning plateaus after roughly 40 trials, so it might be worth trying this analysis on the part where learning took place (e.g., comparing first and second half). Presumably, the monkeys know that one option is rewarded, and the other isn't (i.e. there is an anticorrelation in the reward structure). Are both the chosen and unchosen option updated after each outcome in the modified RL model? If not, would this change any of the conclusions?

(3) To match success rates, the authors inserted forced abort trials in low cost trials. This was done to

match success rates i.e. to decrease the risk associated with HC trials which were more often unsuccessful. But at the same time, it increases the risk associated with LC trials because an abort means that no reward is obtained. It is probably tricky to find an optimal solution but could the higher response to LC vs HC at the time of the cost cue (in both the saccade and uncertain task) be due to a difference in the risk of the trial being aborted, rather than relating to the cost itself? Was a differential response to cost cues also observed in the 'explore' task at the time of the cost cue? These data are currently not reported. And do responses to cost cues differ after an incorrect or an abort trial versus a completed trial? This might provide some insight but if the two interpretations cannot be distinguished, it might be sufficient to mention this possibility of risk or uncertainty (or even 'control') modulating responses to cost cues in the discussion.

(4) It is unclear from Fig 1b how long the monkeys fixated in the low-cost trials, it seems that one monkey voluntarily fixated longer than was necessary. Is it possible to distinguish whether the cost-modulations of the RPE at the time of the reward cue are better explained by the exerted cost which varies on a trial by trial basis, or by the expected/required cost which is fixed for all LC and HC trials? Also, please clarify what proportion of the 2s and 0.5s, respectively, the monkeys had to hold the fixation for in order for the effort to count as successful in the HC and LC trials.

(5) RTs vary as a function of cost and reward at the time of the reward cue. Could this explain some of the firing rate differences observed? It is unclear if RTs were accounted for (i.e., included as confound regressors) in the main analyses.

(6) Some places in the manuscript are phrased in a way that suggests that cost and reward are combined (as a sum or integrated common currency) within the dopaminergic VTA, but by the time the reward cue comes up, the cost has already been paid by the monkeys. While I think this is a very elegant design feature, it needs to be reflected more clearly in some places in the manuscript. It seems one possibility is that a reward obtained after a high cost might be perceived as more rewarding (e.g., because you feel like you have earned/should deserve it), and similarly a zero outcome might be perceived as more disappointing after having put in more work to obtain it. This should be discussed and some of the wording adjusted.

Minor

- How exactly did the authors classify neurons as saliency vs motivational neurons? Were all neurons classified into one of the two categories or just a subset of neurons, and which criteria were used? Currently the details of how these categories were obtained remains unclear.
- The location/depth analysis for saliency vs value neurons seems driven by a few outliers but is not a major part of story (Fig S5a). If it does not hold in a robust regression, I would suggest removing this from the manuscript.
- Why was a different cost (1500ms) used in the exploration task?
- The legend to Fig 7c suggests both monkeys showed an RT effect to reward cues for HC vs LC but this does not seem to be true for monkey P. Please correct.

Reviewer #3:

Remarks to the Author:

In the current study, the authors recorded from putative dopamine neurons in monkey SN and VTA during the performance of several tasks in which the monkeys were required to work to obtain two rewards. One reward was high cost and one was low cost. Consistent with prior work, they found that cues predicting high cost to obtain reward evoked lower RPE-like activity in these neurons. However – critically – the current report also extended these studies dramatically by showing that the same

neurons exhibited higher RPE-like activity to the reward-signaling cues and actual rewards after the work had been done. That is, the dopamine neurons seemed to value the same reward more if the monkey had to work for it than if they did not. This interpretation is supported nicely by behavior of the monkeys and also by an ingenious learning task, in which they show better learning for the high cost cues and rewards. Overall this is an exciting, interesting, and creative study in an area that is full of repetitive and sometimes inscrutable work lately. I really loved it. Indeed I have only minor requests really.

One request is that the authors do more to show how they identify dopamine neurons. Currently they describe criteria in the text, but they do not show this analysis. This has become a very contentious business in the non-primate literature – how to identify dopamine neurons. The sort of criteria used here are often deemed insufficient. Arguing against this silly idea is made more difficult because the primate work does not show the way the waveforms are identified. I would consider it a personal favor if some analysis was presented showing what counts as a dopamine neuron and how it differs from other neuron types. For example, a scatter showing waveform duration versus firing rate or something similar would be extremely helpful.

Related to this, I would also appreciate it if the authors would analyze in supplemental some of the narrow spiking neurons. It would be worthwhile to show that narrow waveform neurons isolated along with the dopamine neurons do not show these correlates.

Lastly I am struck by the failure of the value vs salience distinction to track with the valuation of the high cost versus low cost rewards. I think the authors are 100% correct that if these neurons are coding salience, then they should respond differently from the value neurons. Yet they do not. I think this raises the question of whether these are really coding salience or something else that covaries for the reward and air puff conditions. Would the authors comment on this?

We greatly appreciate the three reviewers for their supportive and helpful comments on our manuscript. We addressed and incorporated all the comments in the revised manuscript. Below we list our responses to each of the reviewers' comments:

Reviewer #1 (Remarks to the Author):

(1) The monkeys made more errors in high-cost trials than in low-cost trials. In order to compensate the success rate (i.e., reward probability), the authors inserted a forced abort in the low-cost trials. This manipulation made the reward prediction error equal between the two cost conditions, and enabled the authors to analyze dopamine neuron activity related to the cost but not the prediction error. This seems to be a clever way to compensate the prediction error. However, I have a concern; if trials in which monkeys have successfully performed are aborted, the monkey's motivation to accomplish the trials would decrease. Such a decrease in motivation affects (probably decreases) the subjective value of reward obtained in the trials. Thus, although the authors reported that the monkeys preferred the reward after high cost and that dopamine neurons were more strongly activated by the reward after high cost, these effects on the monkey's preference and dopamine activation could be explained by the decrease in motivation in the low-cost trials rather than cost itself. This concern must be solved by new analyses or discussed properly.

As we discussed on page 27, the reaction time can reflect valuation of cues. To investigate the effect of forced abort on monkeys' valuation of R cues we examined the relationship between the number of forced abort in the low-cost condition and the reaction time to the R_{HC+} or R_{LC+} cue day by day. However, there were no significant correlations between the number of the forced aborts and the reaction time to the R_{HC+} cue (**Supplementary figure 8a, d**), nor between the number and the reaction time to R_{LC+} cue (**Supplementary figure 8b, e**), nor between the number and the reaction time difference to for R_{HC+} and R_{LC+} cues (**Supplementary figure 8c, f**).

Also we couldn't find significant correlations between the number of the forced aborts in the low-cost condition and the auROC in the DA responses to the R_{HC+}

cue (**Supplementary figure 8g**), between the number and the auROC in the DA responses to the R_{LC+} cue (**Supplementary figure 8h**), nor between the number and the auROC difference for R_{HC+} and the R_{LC+} cues (**Supplementary figure 8i**).

These results imply that the forced aborts had no effects on the monkeys' valuation nor on the difference in activation of the dopamine neurons to the reward cues in the high-cost vs low-cost condition. Therefore, we can say that the forced abort in the low-cost condition could not cause the cost dependent behavioral and neuronal differences to the R cues. We added this description on **page 20**.

(2) In Figure 1e, the authors showed that RT to Rcue is shorter in high-cost trials than in low-cost trials. Based on this data, the authors postulated that the monkeys more preferred the reward in high-cost trials. However, the effect of cost on RT to Rcue is much larger in $R-$ cue than $R+$ cue. This data seems to conflict with the authors' explanation, because the monkey did not obtain the reward after $R-$ cue. If the monkeys really preferred the reward in high-cost trials, the effect of cost on RT has to be larger for $R+$ cue.

We think that a likely explanation for the smaller effect of the cost on the RT for $R+$ compared to $R-$ cues is that the RTs are overall shorter for $R+$ cues than $R-$ cues. Therefore, there is less room for RTs to decrease as a function of cost in the $R+$ condition compared to the $R-$ condition, because of the truncated RT distribution. This is therefore likely an artifact of the overall difference in the RTs between conditions.

On the other hand, why would the monkeys show a shorter reaction time to the R_{HC-} cue than that to the R_{LC-} cue despite the fact that no reward was delivered after the cue presentations? A previous study reported a similar phenomenon (Watanabe et al, 2001). In that study, monkey subjects showed shorter reaction times in unrewarded trials when more preferred rewards were employed in the alternate trials within each block. The authors speculated that the higher overall motivation of the monkeys in the block with more preferred rewards caused a carry-over effect onto the reaction times even for the no reward cue. Similar to that, the expectation of a more valuable reward in the high cost trials might

modulate the reaction time to the no reward cue in the high-cost trials in our task. We added a discussion of this point to page 27.

(3) In the present study, the authors presented the PSTHs of example neurons and the scatter plot for population analyses, but no population PSTH. The population PSTH is more helpful to understand the entire trend of dopamine neuron response, and most of the previous electrophysiological studies on dopamine neurons have shown the population PSTH. The authors need to present it.

We added population PSTHs in Figure 3, 4 and 6.

(4) In the present study, the authors reported that dopamine neurons were suppressed by the start cue. This is unusual. Previous studies have reported that dopamine neurons are activated by cues (e.g., fixation point) indicating trial start. Why did dopamine neurons show the suppression?

The activity of dopamine neurons at the time of onset of a start cue typically signals that a reward can be obtained after the cue. In the present study, however, the subject has to pay an effort cost before obtaining the reward. Because the predicted cost reduces the activity of the dopamine neurons, our dopamine neurons might reduce their activity at the timing of the start cue presentation. We added a reference to this study on page 14.

Reviewer #2 (Remarks to the Author):

(1) The temporal evolution of the encoding of cost and reward information is only shown for representative neurons (and it is not well described how these neurons were chosen). The visualizations of the key results thus rely on a small set of neurons. It would be a lot more convincing if time-course plots could show population average responses: this would provide an intuition about the effect sizes and the peak timing of the effects.

We added population PSTHs in **Figure 3, 4 and 6**.

While the effects seem quite consistent and robust, the way the results are currently plotted, it is hard to be sure and the plots are not very intuitive. The statistics are done on the auROC of all neurons which rely on set time windows in which the signal is evaluated. The choice of these time windows is somewhat unclear. Were they defined based on the animals RT? If so, how was the start point of each window determined?

We defined the time window based on the activity of the dopamine neurons but not the reaction time. The start and end points of time windows were determined based on the rise and fall time of the population averaged response found in previous monkey dopamine studies. We added this detail on **page 9**.

Could significance be reported in sliding windows rather than for one fixed window?

We used a fixed window, because we wanted to minimize the need to correct for multiple comparisons across different test windows. Instead we used an independently defined fixed criterion based on previous studies. The use of a sliding window would have markedly reduced our statistical sensitivity given the need to correct for multiple comparisons across windows.

In addition to that, the timing of the dopamine responses were not constant between the two monkey subjects (**Supplementary fig.2**). If we analyzed the significance in a sliding window, it would be difficult to merge the data from two monkeys.

Furthermore, to compare the dopamine responses to the other experimental data such as reaction time (**Supplementary Fig. 7**) or the number of the forced abort (**Supplementary Fig. 5c, Supplementary Fig. 8g-i**) the dopamine response should be a fixed value rather than the time series data. Therefore, we used a fixed time window to analyze the significance of the effect of the cost in a merged dataset that pooled the data from the two monkeys.

Please justify the choice of Wilcoxon signed-rank tests.

When we examined the effect of the cost on the neuronal data, the numbers of neurons in some neuronal data sets were small (i.e. HLC uncertain task). Therefore, to reduce the effect of outliers, we chose a Wilcoxon's signed-rank test to analyze the neuronal data. We added a justification for this decision on **page 10**.

(2) While the learning effects in the third ('explore') experiment seem robust, several pieces of information are missing. The learning rate α is not currently reported, and it is hard to interpret the values of the critical parameter $wRPE$ without knowing α . Is it also unclear if $\alpha * wRPE$ as the product is restricted to be <1 or whether α can vary between 0-1 and $wRPE$ take any values? In the latter case the product (effective learning rate in HC trials) could become larger than 1 which would make it difficult to interpret (and $wRPE$ goes up to 10 in some cases in Fig 8d which it should only do if the learning rate was very low <0.1).

We did not restrict the range of the parameter $\alpha * wRPE$ between 0-1. Therefore, we re-analyzed the learning effects with restricted learning rate parameters for the two cost conditions (α_{HC} and α_{LC}) in the revised manuscript. We plotted the distribution of the ratio between the two learning rate parameters (α_{HC}/α_{LC}) which is identical to the parameter $wRPE$ in the original manuscript in Figure 8d. As was the case with the analysis presented in the original manuscript, the distribution was significantly larger than zero, therefore indicating that the RPE was significantly larger in the high-cost trials than that in the low-cost trials. We added this description on **page 9 and 24**.

Also, was α fitted on all trials? Fig 8a shows that learning plateaus after roughly 40 trials, so it might be worth trying this analysis on the part where learning took place (e.g., comparing first and second half).

The learning rate was indeed fitted on all trials. Although Fig 8a shows that learning plateaued after roughly 40 trials, the plot is an average – pooling across many individual learning sessions per monkey (49 sessions for monkey P; 86 sessions for monkey S). Individual learning sessions reached asymptote at different rates, for instance some sessions reached asymptote after only 20 trials, while others reached asymptote after as long as 80 trials. Therefore, in our opinion it is not appropriate to use parts of trials for this analysis and all trials are required.

Presumably, the monkeys know that one option is rewarded, and the other isn't (i.e. there is an anticorrelation in the reward structure). Are both the chosen and unchosen option updated after each outcome in the modified RL model? If not, would this change any of the conclusions?

We used a RL model in which only the chosen options are updated for the results in **Figure 8**. To check the reviewer's concern, we also analyzed the learning process with two additional RL models. Both of these models involved updating, the chosen option and unchosen option based on an anticorrelation in the reward structure. One of the models shared the same learning rate parameter for updating the chosen and unchosen option. The other model utilized independent learning rate parameter for chosen and unchosen options. If we used these models to explain the learning process, the RPE was still significantly larger in the high-cost trials than that in the low-cost trials (**Supplementary figure 11**). Thus our conclusions do not change even if we include anti-correlated updates for chosen and unchosen options. We added this information on **page 25**.

(3) To match success rates, the authors inserted forced abort trials in low cost trials. This was done to match success rates i.e. to decrease the risk associated with HC trials which were more often unsuccessful. But at the same time, it increases the risk associated with LC trials because an abort means that no reward is obtained. It is probably tricky to find an optimal solution but could the higher response to LC vs HC at the time of the cost cue (in both the saccade and

uncertain task) be due to a difference in the risk of the trial being aborted, rather than relating to the cost itself?

If the risk of the trial being aborted is increased by the forced abort and caused the preference change and enhanced dopaminergic activation to the LC cue, the number of forced aborts should be related to the preference and the enhanced activation. In other words, as the number of forced abort increases, the monkeys should prefer the LC cue more and the dopamine neurons should respond more to the LC cue. Therefore, we first examined the relationship between the number of the forced abort in the low-cost condition and the difference between the reaction time to the HC cue and the LC cue. However, there were no significant correlations between the number of the forced aborts and the difference between the reaction times (**Supplementary figure 5a-b**). Next, we examined the relationship between the number of the forced aborts in the low-cost condition and the auROC between the dopamine response to the HC cue and the LC cue. If the number of forced aborts increased activation to the LC cue, we would expect to find a negative correlation between the number of forced aborts and the auROC. On the contrary, we found a positive not a negative correlation between the number of the forced aborts and the auROC (**Supplementary figure 5c**). Therefore, we can say that the forced abort in the low-cost condition does not cause the changed preference and enhanced activation to the low-cost cue. We added this description on **page 16**.

Was a differential response to cost cues also observed in the 'explore' task at the time of the cost cue? These data are currently not reported.

Unfortunately, we did not record neuronal activity during the exploration task. The exploration task was implemented as a purely behavioral study. We add clarification on this point to **page 9**.

And do responses to cost cues differ after an incorrect or an abort trial versus a completed trial? This might provide some insight but if the two interpretations cannot be distinguished, it might be sufficient to mention this possibility of risk or

uncertainty (or even 'control') modulating responses to cost cues in the discussion.

We compared the dopamine responses to the cost-cues after abort versus after correct trial (**Supplementary figure 5d**). However, the dopamine responses to the C cues after correct trials were not different from those after abort trials in both cost conditions. These results also support the independence of the modulation of the risk or uncertainty by the forced abort and the cost dependent neuronal modulation. We added a description of these findings to **page 17**.

(4) It is unclear from Fig 1b how long the monkeys fixated in the low-cost trials, it seems that one monkey voluntarily fixated longer than was necessary. Is it possible to distinguish whether the cost-modulations of the RPE at the time of the reward cue are better explained by the exerted cost which varies on a trial by trial basis, or by the expected/required cost which is fixed for all LC and HC trials?

We examined the relationship between the actual fixation durations and the normalized dopamine responses to the reward cues on a trial by trial basis for each cost and reward condition (HC+, HC-, LC+, LC-). However, we could not find any significant correlation between them (**Supplementary figures 7a-d**). These results indicate that the cost dependent modulations of the dopamine response to the reward cue are not explained by the actual fixation duration, yet explained by the expected/required cost which is fixed for all LC and HC trials. We added this description on **page 19**.

Also, please clarify what proportion of the 2s and 0.5s, respectively, the monkeys had to hold the fixation for in order for the effort to count as successful in the HC and LC trials.

During the fixation, if the monkeys moved their gazes beyond a fixation window of $4^\circ \times 4^\circ$, the task was aborted. The fixation window started 400 ms after the fixation point presentation because the monkeys needed time to prepare for the saccade and the adjustment of their fixation. Therefore, in the HLC saccade task,

the monkeys had to fixate for at least 1600 ms in the HC trials and at least 100 ms in the HC or LC trials, respectively. We added this description on page 5.

(5) RTs vary as a function of cost and reward at the time of the reward cue. Could this explain some of the firing rate differences observed? It is unclear if RTs were accounted for (i.e., included as confound regressors) in the main analyses.

Reaction times had not been used for analyzing the cost dependent modulation of the dopamine response in the analysis reported in the original manuscript. To address this, we calculated the correlation coefficient between the reaction times and the normalized dopamine responses to the reward cues on a trial by trial basis for each cost and reward condition (HC+, HC-, LC+, LC-). We could not find any significant correlation between them (**Supplementary figures 7e-h**). This result indicates that the dopamine responses are independent from the reaction times in each trial, yet modulated by the amount of required cost and expected reward which are fixed for each type of trials. We added this result on page 19.

(6) Some places in the manuscript are phrased in a way that suggests that cost and reward are combined (as a sum or integrated common currency) within the dopaminergic VTA, but by the time the reward cue comes up, the cost has already been paid by the monkeys.

In the manuscript, we mention the integration of the cost and reward only at the timing of the cost cue presentation (page 15, page 29).

While I think this is a very elegant design feature, it needs to be reflected more clearly in some places in the manuscript. It seems one possibility is that a reward obtained after a high cost might be perceived as more rewarding (e.g., because you feel like you have earned/should deserve it), and similarly a zero outcome might be perceived as more disappointing after having put in more work to obtain it. This should be discussed and some of the wording adjusted.

We appreciate the reviewer's suggestion. We now discuss this point more clearly on page 30.

Minor

- How exactly did the authors classify neurons as saliency vs motivational neurons? Were all neurons classified into one of the two categories or just a subset of neurons, and which criteria were used? Currently the details of how these categories were obtained remains unclear.

We classified all dopamine neurons into one of the two categories based on their response to the air-puff stimuli. If the response of a neuron to the air-puff stimuli was smaller than the spontaneous activity level, then the neuron was classified as being of the motivational value type (**Figure 3e**). If the response of a neuron to the air-puff stimuli was larger than the spontaneous activity level, then the neuron was classified as being of the salience type (**Figure 3h**). We added these details to page 10.

- The location/depth analysis for saliency vs value neurons seems driven by a few outliers but is not a major part of story (Fig S5a). If it does not hold in a robust regression, I would suggest removing this from the manuscript.

If we calculated the relationship without two outliers, the significant difference between the recording locations of salience and value type neurons indeed disappeared. Therefore, according to the reviewer's suggestion, we removed these results from the manuscript.

- Why was a different cost (1500ms) used in the exploration task?

We found that if the monkeys performed the exploration task with a 2000 ms fixation duration, they performed the task with a very low success rate perhaps because of the difficulty of maintaining fixation for that duration or because of a very low reward rate. Therefore, to reduce the difficulties of the task and

increase the success rate, we used a different cost (1500 ms) for the exploration task. We added this description on page 7.

- The legend to Fig 7c suggests both monkeys showed an RT effect to reward cues for HC vs LC but this does not seem to be true for monkey P. Please correct.

Thank you for spotting this incorrect description. We corrected this.

Reviewer #3 (Remarks to the Author):

One request is that the authors do more to show how they identify dopamine neurons. Currently they describe criteria in the text, but they do not show this analysis. This has become a very contentious business in the non-primate literature – how to identify dopamine neurons. The sort of criteria used here are often deemed insufficient. Arguing against this silly idea is made more difficult because the primate work does not show the way the waveforms are identified. I would consider it a personal favor if some analysis was presented showing what counts as a dopamine neuron and how it differs from other neuron types. For example, a scatter showing waveform duration versus firing rate or something similar would be extremely helpful.

After looking again in detail at our method for identifying dopamine neurons in the light of the reviewer's comment, we have concluded that the threshold we had used in the analysis presented in the original manuscript to separate dopamine from non-dopamine neurons to produce the results presented in the original manuscript was actually too liberal, in that a subset of neurons with relatively short spike waveforms and high spontaneous firing rates had been included in the original analyses. In the present version of the manuscript we report a revised analysis with a more stringent exclusion criterion in which we excluded neurons with short spike waveforms (<300 \$\mu\$ s) and high spontaneous firing rates (>6 Hz). As we consider our new thresholding approach to be more appropriate compared to the original approach we had used, we replaced all of the figures with results from this new analysis. As a result, the number of the

neurons included in the neuronal data analyses were decreased in the revised manuscript (from 83 to 70 in the HLC saccade task; from 24 to 19 in the HLC uncertain task). However, essentially all the results remain unchanged with regard to the statistical evidence supporting our conclusions, except for one result in supplementary figure 4c. In the original manuscript, the response to the HC cue relative to the spontaneous activity was reported as being significantly higher than zero, but the response to the HC cue no longer exhibits significant activation in the revised manuscript. However, the change in this result is minor and does not affect the overall conclusion.

We add a scatter plot which shows the waveform duration versus spontaneous firing rate in **Supplementary figure 3b**.

Related to this, I would also appreciate it if the authors would analyze in supplemental some of the narrow spiking neurons. It would be worthwhile to show that narrow waveform neurons isolated along with the dopamine neurons do not show these correlates.

We skipped recording neuronal activities from putative non-dopaminergic neurons that showed a narrow waveform to increase our chance to record from more promising DA neuron candidates. Therefore, we cannot report the activity profile of the narrow waveform neurons as there are too few neurons in that category in the final dataset, unfortunately.

Lastly, I am struck by the failure of the value vs salience distinction to track with the valuation of the high cost versus low cost rewards. I think the authors are 100% correct that if these neurons are coding salience, then they should respond differently from the value neurons. Yet they do not. I think this raises the question of whether these are really coding salience or something else that covaries for the reward and air puff conditions. Would the authors comment on this?

In the present study, the salience type dopamine neurons, which showed phasic activation to the unpredictable aversive stimulus, did not show extra activation to

the high-cost predicting cue. One possible explanation for this inconsistency is that the cost or effort in our experiments were not salient enough to induce the response of the salience type dopamine neurons. Here, we categorized the value and the salience type dopamine neurons based on the response to the water reward and the aversive air-puff stimulus. When the reward or the air-puff were delivered, the monkeys could perceive them in a moment. Therefore, the reward and the air-puff deliveries were the “salient” events for the monkeys. On the other hand, when they paid the cost it required a few second to find the end of the cost because they performed the fixation. Therefore, the cost in our experiments were “not salient” and the salience type dopamine neurons might not show activation to the cost cues.

The other possibility is that the salience type dopamine neurons respond to events after which some movements to avoid aversive stimuli were induced. When the air-puff was delivered to the monkeys, they make some movements such as eye blink. However, in the HLC saccade task, the monkeys had to make a saccade to the fixation target and to keep their gaze on the fixation target without any movement as the cost. Actually, a recent study showed that dopamine release in the nucleus accumbens following a reward-predicting cue was attenuated unless movement was correctly initiated. Therefore, the cost in our experiments did not initiate any movement and that might result in the inconsistent response of the salience type dopamine neurons.

We added these discussions on **page 31**.

Reviewers' Comments:

Reviewer #1:

Remarks to the Author:

The authors have appropriately addressed all the concerns I raised previously, and have improved the manuscript to the level suitable for publication in Nature Communications.

Reviewer #2:

Remarks to the Author:

The authors have done a great job at addressing my comments. I have only a few minor outstanding points:

(1) Thanks for showing population responses in the updated figures. These should incorporate error bars.

(2) Were time windows determined based on previous monkey studies or on the average population response in this study? The response to the reviewers seems to suggest previous studies but that does not match the manuscript text.

(3) As far as I can see, the authors still only report the ratio of the learning rates and softmax inverse temperature parameters for HC and LC, giving no insight into the actual range of alpha values found. It would be helpful if raw alpha values could be reported at least as mean and std or in a table.

Reviewer #3:

Remarks to the Author:

Fantastic paper. Thanks for addressing my comments.

We greatly appreciate the three reviewers for their supportive and helpful comments on our manuscript. We addressed and incorporated all the comments in the revised manuscript. Below we list our responses to each of the reviewer's comments:

****REVIEWERS' COMMENTS:**

Reviewer #1 (Remarks to the Author):

The authors have appropriately addressed all the concerns I raised previously, and have improved the manuscript to the level suitable for publication in Nature Communications.

Reviewer #2 (Remarks to the Author):

(1) Thanks for showing population responses in the updated figures. These should incorporate error bars.

We added SEMs on population PSTHs in **Figure 3, 4 and 6**.

(2) Were time windows determined based on previous monkey studies or on the average population response in this study? The response to the reviewers seems to suggest previous studies but that does not match the manuscript text.

Thank you for spotting this incorrect description. We determined the time windows based on the population averaged response using previous monkey dopamine studies as references. We corrected this description.

(3) As far as I can see, the authors still only report the ratio of the learning rates and softmax inverse temperature parameters for HC and LC, giving no insight

into the actual range of alpha values found. It would be helpful if raw alpha values could be reported at least as mean and std or in a table.

We added plots about the distribution of the fitting parameters (a , b , α and β) in **Supplementary figure 11 and 12**.

Reviewer #3 (Remarks to the Author):

Fantastic paper. Thanks for addressing my comments.